# High Failure Rates of Locking Compression Plate Osteosynthesis with Transverse Fracture around a Well-Fixed Stem Tip for Periprosthetic Femoral Fracture

**DOI:** 10.3390/jcm9113758

**Published:** 2020-11-22

**Authors:** Byung-Woo Min, Kyung-Jae Lee, Chul-Hyun Cho, In-Gyu Lee, Beom-Soo Kim

**Affiliations:** Department of Orthopedic Surgery, Keimyung University Dongsan Hospital, Keimyung University School of Medicine, 1035 Dalgubeol-daero, Dalseo-gu, Daegu 42601, Korea; min@dsmc.or.kr (B.-W.M.); oslee@dsmc.or.kr (K.-J.L.); oscho@dsmc.or.kr (C.-H.C.); 180227@dsmc.or (I.-G.L.)

**Keywords:** periprosthetic femoral fracture, locking compression plate, osteosynthesis, open reduction, minimally invasive, complications, failure

## Abstract

This study investigated the incidence of failure after locking compression plate (LCP) osteosynthesis around a well-fixed stem of periprosthetic femoral fractures (PFFs). We retrospectively evaluated outcomes of 63 Vancouver type B1 and C PFFs treated with LCP between May 2001 and February 2018. The mean follow-up duration was 47 months. Only patients with fracture fixation with a locking plate without supplemental allograft struts were included. We identified six periprosthetic fractures of proximal Vancouver B1 fractures with spiral pattern (Group A). Vancouver B1 fractures around the stem tip were grouped into seven transverse fracture patterns (Group B) and 38 other fracture patterns such as comminuted, oblique, or spiral (Group C). Vancouver C fractures comprised 12 periprosthetic fractures with spiral, comminuted, or oblique patterns (Group D). Fracture healing without complications was achieved in all six cases in Group A, 4/7 (57%) in Group B, 35/38 (92%) in Group C, and 11/12 (92%) in Group D, respectively. The failure rates of transverse Vancouver type B1 PFFs around the stem tip were significantly different from those of Vancouver type B1/C PFFs with other patterns. For fracture with transverse pattern around the stem tip, additional fixation is necessary because LCP osteosynthesis has high failure rates.

## 1. Introduction

Periprosthetic femoral fractures (PFFs) remain some of the most challenging complications following hip joint replacements. In addition, their incidence is projected to rise with the increasing use of hip replacements and the higher risk of falls associated with an aging population [1,2,3,4]. The principal treatment modality for Vancouver type-B1 and C fractures is open reduction and internal fixation (ORIF). However, surgeons have not yet reached a consensus on the optimal method for reduction and fixation [3,5,6]. Although locking compression plate (LCP) osteosynthesis is considered reliable to manage periprosthetic fractures using a well-fixed stem, previous studies have presented evidence of the high failure rate of LCP fixation [3,5,7,8]. Thus, this study aimed to document the incidence of LCP osteosynthesis failure around a well-fixed stem in PFFs.

## 2. Materials and Methods

We retrospectively identified a consecutive series of 72 patients with Vancouver type B1 and C PFFs treated with LCP between May 2001 and February 2018. Only patients with fracture fixation with an LCP (Synthes, Paoli, PA) without supplemental allograft struts or dual plate fixation were included in the study. Intraoperative PFFs and atypical fracture patterns were excluded. We followed these patients up at scheduled visits to evaluate radiographic and clinical outcomes. Of the 72 patients considered for this study, four patients were lost to follow-up monitoring before the end of the minimum one-year follow-up period, and five patients died due to unrelated reasons. Finally, 63 patients were enrolled in the study (Figure 1).

The PFFs were classified according to the Vancouver classification system [9,10]. Vancouver B1 fractures were subdivided according to location, fractures of the proximal part of the stem, and fractures around the tip of the stem. We defined the fractures around the tip as those cases with the primary fracture line passing the tip of the stem. Classification of the fractures according to the fracture pattern were as follows: oblique, spiral, transverse, and comminuted fractures. We identified six periprosthetic fractures of proximal Vancouver B1 fractures (Group A). Vancouver B1 fractures around the stem tip were grouped into seven transverse fracture patterns (Group B) and 38 other fracture patterns, such as comminuted, oblique, or spiral (Group C). Vancouver C fractures consisted of 12 periprosthetic fractures with spiral, comminuted, or oblique patterns (Group D) (Figure 1). Of these 63 patients, 36 and 27 were male and female, respectively. The mean age was 67.6 years (range: 32–91 years) and the mean body mass index was 22.1 (range: 15.6–32.0) at the time of surgery. The mean follow-up period was 47 months (range: 12.0–168 months). We retrospectively compared the outcomes of patients in each group (Table 1). Informed consent was obtained from all participants, all research was performed in accordance with the guidelines and regulations of the Journal, and Institutional Review Board approved the protocol (DSMC 2019-03-029).

### 2.1. Surgical Procedure

All procedures were performed at one institution by the same surgeon using a lateral approach. No intraoperative prosthetic hip joint dislocations occurred. The differential diagnosis between Vancouver types B1 and B2 was based on prosthetic loosening and fracture patterns observed on plain radiographs and computed tomography images. Additionally, cases where the distinction between B1 and B2 fractures was preoperatively not clear were excluded from this study.

Surgery was performed according to the location and pattern of the fracture using minimally invasive plate osteosynthesis (MIPO) in 32 patients, and ORIF in 31 patients. Further, the patients who had Vancouver B1 fractures were operated on an orthopedic fracture table and traction was applied through a boot; patients with Vancouver C fractures were operated on a radiolucent table. Two incisions were made in the proximal and distal femur along the marking line while excluding the fracture site in case of MIPO technique. LCP was slipped into the submuscular supra-periosteal position and reduction in the fracture site was attempted indirectly by fluoroscopically guided external maneuvering to include less damage to the soft tissue and preserve periosteal circulation in the fractured bone [11]. While maintaining the reduction status, distal and proximal positioning screws were fixed. To avoid the stress build-up between the prosthesis and the LCP, both the devices were overlapped. For reducing the risk of fixation failure, only 3–4 screws were inserted in each fragment to maintain proper distance between the stress centers on the plate. This aided in reducing the stress while pulling out screws and distributed the stress on the center of the plate at the fracture site [5,6]. Additional fixation was carried out using a locking attachment plate (Synthes, Oberdorf, Switzerland) or cables when bicortical screw fixation was impossible due to proximal femoral prosthesis (Figure 2).

The direct lateral approach was used for the plate fixation in the case of the ORIF technique depending on the fracture location. The iliotibial band was incised, and the vastus lateralis was elevated from the posterior fascia. The periosteal exposure was limited to the fracture only to allow reduction. The plate was directly fixed to the bone after anatomical reduction.

The postoperative rehabilitation was the same for all patients, who were allowed partial weight-bearing during callus appearance around the fracture area on follow-up radiographs. All patients received 24-h antibiotic prophylaxis and thromboembolic prophylaxis with low-molecular-weight heparin for 30 days. No patient received prophylaxis against heterotopic ossification.

### 2.2. Outcome Evaluation

For comparative analysis across the four groups, the surgical success rate was set as the primary outcome. The success rate was defined as the fracture healing without complications. Complications were confirmed if the patients developed at least one of the previously cited complications: non-unions, malunions, hardware failure, infection, or reoperations [3]. Any discrepancy concerning the outcome data was resolved by consensus between the two authors (KJL, BSK). The “Non-union” classification included pertinent cases of non-union reported by the author or a case of repeat surgical fixation because no union occurred after the initial fixation. “Malunion” was defined as deviation from anatomic alignment > 5 degrees in any plane. “Hardware failure” included plate fracture/pull-out; isolated fractures of screws or cerclage wires that did not affect the clinical outcome were excluded. “Infections” included deep infections requiring intravenous antibiotics or surgical irrigation and debridement. “Reoperation” was defined as cases requiring at least one reoperation (such as irrigation/debridement, revision arthroplasty, revision fixation, or hardware removal) [3].

Other assessments were based on the number of bone union formations, bone union formation time, operative time, blood loss during surgery, number of blood transfusions after surgery, and complications developed by the four groups.

### 2.3. Statistical Analysis

We analyzed the differences in the demographic parameters by chi-square tests and Mann–Whitney tests. We tested the assumption of normal distribution and homogeneity of variance using the independent sample t-test, Kolmogorov–Smirnov test, and Levene’s test. The level of significance was set at *p* < 0.05. The SPSS statistical package (version 22.0; IBM, Armonk, NY, USA) was used for the analysis and modeling of the data.

## 3. Results

Fracture healing without complications was achieved in 56 of 63 patients (89%) in the average time period of 5.9 months (range: 5–8 months). Among the four groups, all cases in Group A, 4 cases out of 7 cases (57%) in Group B, 35 cases out of 38 (92%) in Group C, and 11 cases out of 12 cases (92%) in Group D had successful primary outcomes (Table 2). There were significant differences in the incidence of failure between Vancouver type B1 PFFs around the stem tip with a transverse fracture pattern and other Vancouver type B1/C PFFs with other fracture patterns (*p* < 0.001).

Seven cases of failure in 63 patients occurred during the follow-up period due to non-union, malunions, hardware failure, infection, or reoperations. Kaplan–Meier survival analysis comparing the four groups with the endpoint as complications developing due to any reason showed significant differences between the groups (log-rank test *p* = 0.010) (Figure 3).

No significant differences were observed except in Group B for the different stems (cemented versus uncemented) that were retained before fractures occurred. However, in Group B, the surgical success rates with the uncemented stems were significantly lower (*p* < 0.001) (Table 3). There was no difference in the success rate except for Group B when the stem was divided according to the number of surgeries. In the case of Group B, treatment failed in both patients with revisional stem surgery (Table 4). In the comparison between the surgical methods (ORIF versus MIPO), there was no difference in all groups, and in Group B, both methods had a high failure rate (Table 5).

The median time for union formation was 5.9 months (range: 5–8) with no differences among the four groups (*p* = 0.586). The intraoperative blood loss and number of postoperative blood transfusions were not different among the groups. However, Group A had a significantly shorter operative time than other groups did (*p* = 0.049) (Table 6).

Three failure cases with metal breakage caused by non-union were noted in Group B (Figure 4). In the other groups, two cases of stem subsidence with fracture healing and one case of deep infection occurred in Group C; one case of metal failure with non-union occurred in Group D due to inappropriate fixation. All complications were treated by revisional surgery (Table 7).

## 4. Discussion

This study primarily evaluated the incidence of LCP osteosynthesis failure around a well-fixed stem in periprosthetic femoral shaft fractures. Empirically, the failure rate was higher for the simple transverse fracture pattern around the stem tip than for another simple or comminuted fracture pattern. The findings of this study demonstrated that Vancouver type B1 PFFs around the stem tip with a transverse fracture pattern showed a higher incidence of failure. The results of this study present preliminary evidence contributing to the efforts to reach consensus surrounding the management of Vancouver type B1 PFFs with a transverse fracture pattern.

Vancouver type B1 and C fractures are treated using ORIF or MIPO due to stable maintenance of the femoral stem, but there is no consensus among surgeons on the optimal reduction and fixation method [3,4,12,13,14,15]. Although LCP osteosynthesis has been considered reliable for managing periprosthetic fractures with a well-fixed stem [16,17,18,19], some authors presented evidence of high failure rate of LCP fixation [3,5,7,8]. Ricci et al. [19] reported successful healing and no malunions in 50 Vancouver type B1 periprosthetic fractures treated by indirect fracture reduction and fixation using a cable plate without allograft. However, Buttaro et al. [5] showed a high failure rate when using only LCPs for these fractures. In our study, 56/63 patients (89%) had good results, as with previous studies. These results show that fixation with LCP for Vancouver type B1 and C fractures is effective for the treatment of fractures. However, when the seven failed cases were analyzed, the failure rate was high when the fracture pattern was a transverse feature, although dynamization was achieved using the combi hole of the plate [20]. Therefore, this study was performed to classify fractures according to their patterns and analyze them. Our results were statistically significant, with a higher failure rate in the case of a transverse fracture pattern around the stem tip. In addition, except for the patients who had the transverse fracture pattern, and one case of misdiagnosis where stem subsidence occurred, 52/54 patients (96%) had successful results that indicated the efficacy of LCP without any additional implants. Therefore, even in B1 fractures, which require osteosynthesis, the treatment method should be chosen by considering the fracture pattern.

Previous studies focused mainly on ORIF with or without allogeneic strut graft or revision arthroplasty procedures. Recently, however, Chakrabarti et al. [21] reported that the success rate for Vancouver B1 fractures of PFFs depends on the fracture pattern. They reported that eight fractures in 32 patients with transverse or short oblique patterns were treated with ORIF with cemented stems using cable-ready bone plates, and 50% of these had non-union. Thus, they insisted that revision arthroplasty may be preferable, otherwise patients may require supplementary fixation with cortical onlay graft. Contrary to their study, our study had more subjects with different prosthesis (only LCP) used for surgery. In contrast, the patients who failed in their study were those who received cemented stems, and the patients who failed in our study were those who received uncemented stems. Therefore, the failure rate of fractures with transverse patterns regardless of stem type is extremely high, warranting extra vigilance during the treatment.

In terms of the treatment principles of fracture management, in the case of a transverse fracture pattern in general fractures, compression of the fractured site is the most important factor in the bone healing process [20]. Muller et al. reported good results for plate fixation using dynamization in transverse/short oblique fractures, [22] but this does not provide adequate compression. Applying proper compression is a challenge due to a stem in case of periprosthetic fractures that inhibits fracture reduction, and proper compression is needed to violate the periosteum around the fracture line for fracture with a transverse pattern, which interferes with callus generation. To overcome this problem, previous studies have reported good results using strut allografts or dual plates with LCP osteosynthesis to increase the stability of fracture fixation [4,6,12,23]. Three patients with complications in this study were treated and achieved bone union by using strut onlay allograft. Therefore, when devising a method for the treatment of Vancouver type B1 or C fractures, treatment of LCP may be possible only if the fracture had a conventional pattern, but ORIF using an additional implant or fixation method is necessary in cases of transverse fracture patterns.

The limitations of this study included the small size of the cohort (63 patients, 63 hips) and the shorter-term follow-up after surgery (an average of 47 months). Additionally, we focused on the bony healing and fracture-related complications instead of the implant survival. Moreover, because some type of fracture pattern is rare, there is a large difference in the number of patients by group. In addition, the limitation between B1 and B2 fracture is that some cases may be confused due to the situation in which the distinction between B1 fracture and B2 fracture should be performed before surgery. Despite these limitations, the findings are of value because this study includes all cases of Vancouver type B1 and C fractures treated by a single surgeon (BWM) using only LCP since 2001 at one institution. Moreover, we studied the results of a uniform treatment method and implant (only LCP) according to various fracture types, and the results were analyzed in detail by comparing the results according to the fracture pattern.

## 5. Conclusions

In conclusion, osteosynthesis using LCP satisfactorily treats Vancouver type B1 and C PFFs. However, in the case of transverse Vancouver type B1 PFFs around the stem tip, an additional fixation method is necessary because osteosynthesis using LCP alone has a high failure rate.

## Figures and Tables

**Figure 1 jcm-09-03758-f001:**
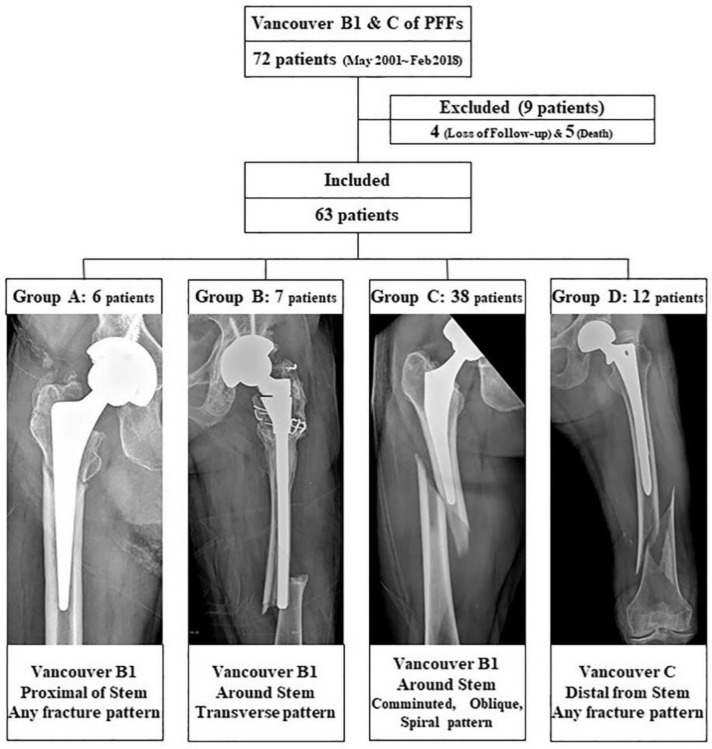
Patient profiles and the groups included in the study.

**Figure 2 jcm-09-03758-f002:**
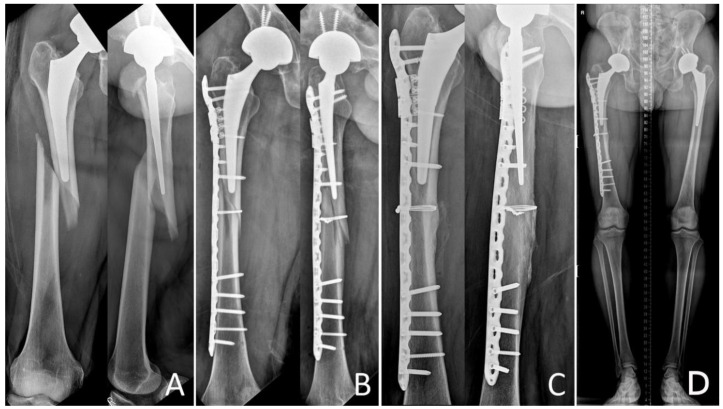
A 46-year-old man underwent total hip arthroplasty because of osteonecrosis of the right femoral head. (**A**) Radiographs show a Vancouver type B1 periprosthetic femoral fracture with a spiral pattern around the stem tip area. (**B**) Radiograph obtained immediately after fixation with the locking compression plate with locking attached plate by the minimally invasive plate osteosynthesis technique. (**C**) Radiograph obtained eight months after fixation shows union of the Vancouver type B1 periprosthetic femoral fracture. (**D**) The lower limb scan image obtained eight months after fixation shows bone healing with appropriate alignment.

**Figure 3 jcm-09-03758-f003:**
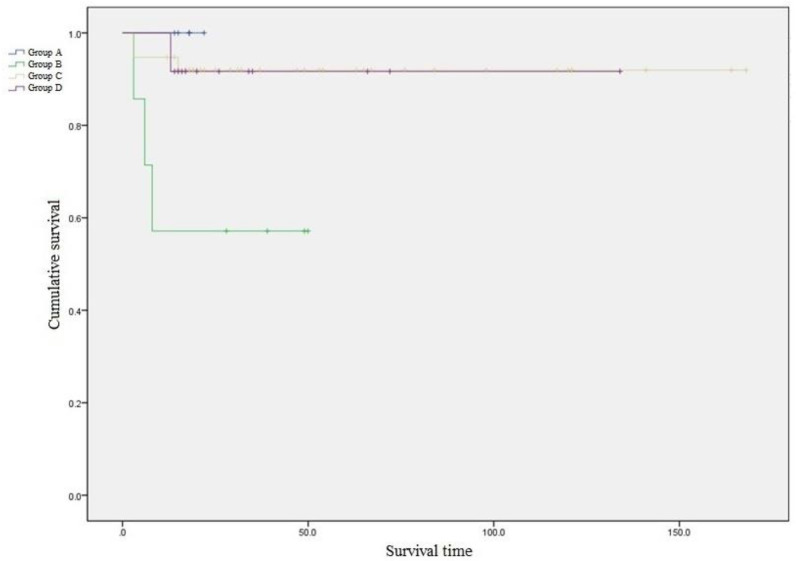
Kaplan–Meier survival analysis comparing the four groups (with the endpoint as the development of complications) showed a significant difference between the groups.

**Figure 4 jcm-09-03758-f004:**
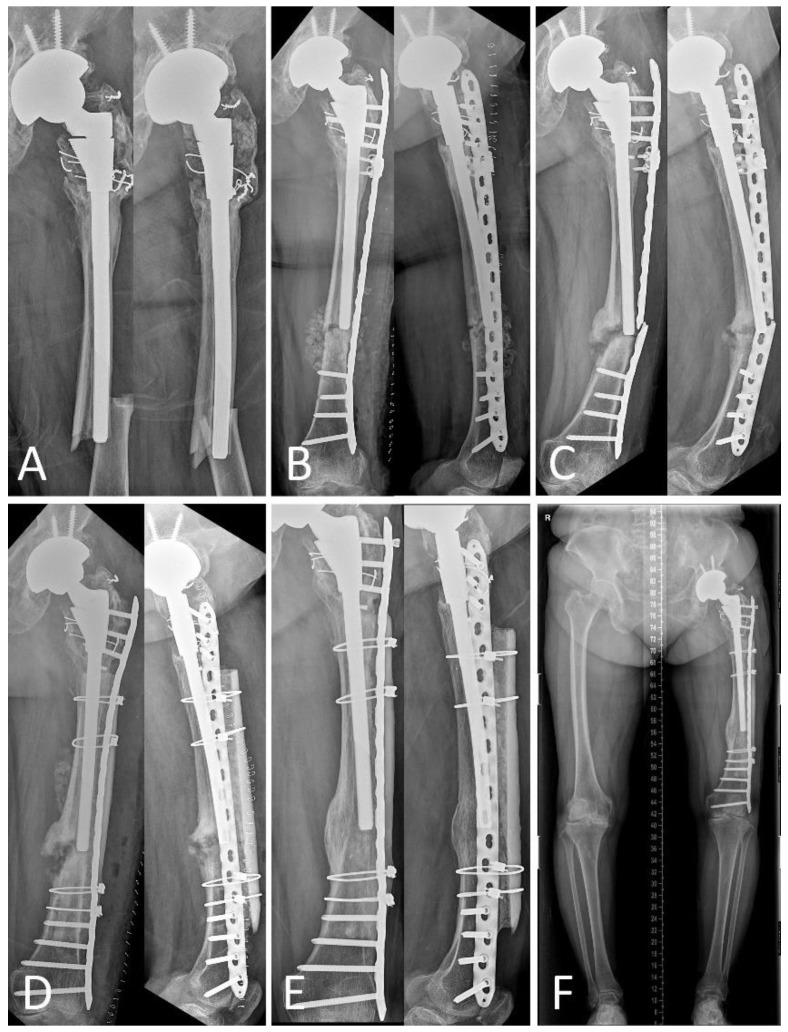
A 72-year-old woman underwent total hip arthroplasty because of left hip osteoarthritis. (**A**) Radiographs show Vancouver type B1 periprosthetic femoral fracture with a transverse pattern around the stem tip. (**B**) Radiograph obtained immediately after fixation with a locking compression plate with locking attached. (**C**) Radiograph obtained 11 months after fixation shows non-union of the fracture site with metal failure. (**D**) Radiograph obtained after revisional surgery using a locking compression plate with strut onlay allograft. (**E**) Radiograph obtained 22 months after revision shows well-healed bone at the fracture site. (**F**) The lower limb scan image obtained 22 months shows a similar alignment to the healthy side.

**Table 1 jcm-09-03758-t001:** Baseline data of patients.

	Group A	Group B	Group C	Group D	Total	*p* Value
Index Age	70.3 (55–82)	73.0 (59–85)	66.4 (32–85)	66.5 (38–91)	67.6 (32–91)	0.352
Gender (M:F)	3:03	1:06	22:16	10:02	36:27:00	0.274
Laterality (L:R)	4:02	4:03	21:17	5:07	34:29:00	0.311
Body mass index (kg/m^2^)	21.6 (15.6–32.0)	22.7 (16.1–26.6)	22.6 (15.6–29.5)	20.5 (18.9–25.5)	22.1 (15.6–32.0)	0.345
Follow-up (month)	17 (14–22)	34 (17–50)	57 (12–168)	39 (14–134)	47 (12–168)	0.298

Group A: Vancouver type B, which shows fracture at the proximal stem; Group B: Vancouver type B, which shows transverse fracture around the stem tip; Group C: Vancouver type B, which shows comminution, oblique, or spiral fracture around the stem tip; Group D: Vancouver type C fracture.

**Table 2 jcm-09-03758-t002:** Surgery success rates of locking compression plate fixation after periprosthetic femoral fractures in each group.

Group	Total	Success	Failure	Success Rate (%)	*p*-Value
A	6	6	0	100	<0.001 *
B	7	4	3	57	
C	38	35	3	92	
D	12	11	1	92	
Total	63	56	7	89	

*: Significant.

**Table 3 jcm-09-03758-t003:** Success rates of fixation according with type of stem.

Group		Total	Success	Success Rate (%)
A	Cemented	0	0	-
	Uncemented	6	6	100
B	Cemented	3	3	100
	Uncemented	4	1	25
C	Cemented	1	1	100
	Uncemented	37	34	92
D	Cemented	1	1	100
	Uncemented	11	10	91

**Table 4 jcm-09-03758-t004:** Success rates of fixation according to primary or revisional stem after hip joint replacement.

Group		Total	Success	Success Rate (%)
A	Primary stem	6	6	100
	Revisional stem	0	0	-
B	Primary stem	5	4	80
	Revisional stem	2	0	0
C	Primary stem	23	21	91
	Revisional stem	15	14	93
D	Primary stem	10	9	90
	Revisional stem	2	2	100

**Table 5 jcm-09-03758-t005:** Success rates of fixation according to surgical methods.

Group		Total	Success	Success Rate (%)
A	ORIF	0	0	-
	MIPO	6	6	100
B	ORIF	4	2	50
	MIPO	3	1	33
C	ORIF	24	22	92
	MIPO	14	13	93
D	ORIF	2	2	100
	MIPO	10	9	90

ORIF: open reduction and internal fixation; MIPO: minimally invasive plate osteosynthesis.

**Table 6 jcm-09-03758-t006:** Postoperative outcomes of each group.

	Group A	Group B	Group C	Group D	*p*-Value
Fracture healing time (months)	5.7	6.3	6	5.6	0.586
Operation time (min)	191	280	232	225	0.050 *
Blood loss during operation (cc)	283	348	344	249	0.766
Transfusion after operation (pack)	0.3	0.7	1.0	0.5	0.953

*: Significant.

**Table 7 jcm-09-03758-t007:** Revisional surgery for patients with each complication after fixation of periprosthetic femoral fractures.

Group	Complication	Period (Months)	Treatment
B	Hardware failure	6	Re-fixation with strut allograft
B	Hardware failure with non-union	8	Re-fixation with strut allograft
B	Hardware failure	3	Re-fixation with strut allograft
C	Stem subsidence	3	Stem revisional operation
C	Stem subsidence	3	Stem revisional operation
C	Infection	15	I&D (PROSTALAC)
D	Hardware failure with non-union	13	Re-fixation with Locking Attachment Plate

Group B: Vancouver type B, which shows transverse fracture around the stem tip; Group C: Vancouver type B comminution, oblique, or spiral fracture around the stem tip; Group D: Vancouver type C fracture.

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
