# Peer review of "High Failure Rates of Locking Compression Plate Osteosynthesis with Transverse Fracture around a Well-Fixed Stem Tip for Periprosthetic Femoral Fracture"

_jcm, 2020, doi:10.3390/jcm9113758_

Round 1

Reviewer 1 Report

Important paper adressing the treatment of periprosthetic fractures of the femur by plating (LCP). One aspect is especially important and worth publishing: Short, transverse fractures at the stem tip show a high failure rate - in these cases adittional stabilisation (allograft or adittional plate) should be considered....

Abstract: Adequate

Introduction:

Short and straight forward... A paragraph outlining the special problems of transverse fractures at the stem-tip could be of interest.

M and M:

Grouping: As the Vancouver classification uses A, B and C, maybe the groups of your study should not use capitals.. sometimes confusing...

Surgical Proc: Did you assess intraoperatively whether the stem was loose? As a significant proportion of fractures praeop. classified as B1 (even with experienece) turn out to be actually B2 (e.g. Lindahl et al.), I think this is an important measure to decide on how to treat the patient. In that context some authors think that a revision of the hip including dislocation is mandatory. I would not believe that this should be done in every case as stability of the implant can be assessed via the fracture - though eavluation of implant-stability in a minimally invasive setting might be challenging. Please comment..

Results:

Table 3: Why did cemented stems perform better in group B? Any idea?

Table 5: Two cases of stem subsidence in group C... Could it be the case, that these fractures were actually B2 and should have been treated by stem revision initially?

Discussion:

Undouptfully, the LCP is a nice device to fix periprosthetic fractures. But - due to it´s delicate design - we, like others, experienced hardware failure especially in difficult situations. As your results and experiences were consistently good, maybe this fact could be linked to the relatively low BMI in your population (lucky guys! I wish I could change at least some of my patients..)? Please discuss...

Lines 216 - 226: Confusing... please restructure

Conclusion: Adequate

Author Response

Dear. Reviewer 1

Thank you for your sincere response.

According to your comments, we did our best to revise our paper. We believe that this process makes our study more valuable. Our answers to your comments are below.

<Response to Reviewer 1 Comments>

M and M:

Grouping: As the Vancouver classification uses A, B and C, maybe the groups of your study should not use capitals.. sometimes confusing...

-> Thanks for your comment. I am sorry that I cannot make any corrections on this part. When dividing the group, it was inevitable because it was divided into 3 groups in the Vancouver type B1, and I think it is more natural to use Capitals rather than lowercase letters in the group name.

Surgical Proc: Did you assess intraoperatively whether the stem was loose? As a significant proportion of fractures preop. classified as B1 (even with experienece) turn out to be actually B2 (e.g. Lindahl et al.), I think this is an important measure to decide on how to treat the patient. In that context some authors think that a revision of the hip including dislocation is mandatory. I would not believe that this should be done in every case as stability of the implant can be assessed via the fracture - though evaluation of implant-stability in a minimally invasive setting might be challenging. Please comment..

-> Thanks for your comment. We generally do not judge B1 and B2 during surgery. Because the direction of treatment is completely different, it is a principle to judge the two types before surgery. As you mentioned, I think there are some cases where there is no clear distinction between B1 and B2. In this study, stem subsidence of 2 patients occurred in group C, and the results were not different even if these cases were excluded. And Cases where the distinction between B1 and B2 fractures was not clear were excluded from this study. I will add more information on this part.

Table 3: Why did cemented stems perform better in group B? Any idea?

-> Thanks for your comment. No clear cause has been identified. This is simply an observed result. It was true, but not so remarkable, that the reduction tended to be easier in transverse fractures around the stem tip when the cement stem was used. I think more cases need to be gathered to answer your questions.

Table 5: Two cases of stem subsidence in group C... Could it be the case, that these fractures were actually B2 and should have been treated by stem revision initially?

-> Thanks for your comment. You might be right. However, all these patients developed from uncemented stems, and at the time of the fracture, they were examined in various aspects on examination, but decided that it was B1 fracture rather than B2. In the case of a cementless femoral stem, we have already identified the type of femoral stem and the fracture line of the femur fracture invaded the area where the femoral stem was fixed or fractured around the cover of stem. We have grasped all these parts and judged that there was no such instability.

Discussion:

Undouptfully, the LCP is a nice device to fix periprosthetic fractures. But - due to it´s delicate design - we, like others, experienced hardware failure especially in difficult situations. As your results and experiences were consistently good, maybe this fact could be linked to the relatively low BMI in your population (lucky guys! I wish I could change at least some of my patients..)? Please discuss...

-> Thanks for your comment. First of all, there was no difference in BMI between the four groups. Koreans, the subject of our study, have the 33rd lowest BMI out of 34 OECD countries. I think we are lucky too. As for BMI, it is not a meaningful result between each group and the success and failure groups, so I think it is unfortunately not appropriate to discuss in the Discussion section.

Lines 216 - 226: Confusing... please restructure

-> Thanks for your comment. We revised that. Please confirm that.

Reviewer 2 Report

Interesting paper concerning the differences in outcomes between patients undergoing ORIF for B1 and C periprosthetic femoral fractures of a different location and fracture pattern in a retrospective comparative study.

The paper's strength is that a single surgeon treated all cases with the same implant and the comparative data.

However, the findings of this study and their interpretation could be misleading, considering that there are potential weaknesses in the methodology of the study. The following are several concerns related primarily to the methodology and results that need to be addressed before a proper interpretation of the findings:

  1. This is a retrospective study with minimal evidence.

  1. How was the sample size of patients calculated? Is this sample size enough to compare the groups and generalize the results? It is not clear. If the sample is not the appropriate one, the results are invalid. Did the number of patients comparable between groups? (6, 7,36 and 12)

  1. Did all the patients undergo a primary uneventful total hip arthroplasty before periprosthetic fracture? Did all the patients receive the same or at least primary stems?It seems to me that the primary THA in the patient of group B in Figure 1 was not uneventful (intraoperative fracture? / this is not a primary stem/ proximal osteolysis)

  1. In my opinion, the paper's main disadvantage is the differential diagnosis between B1 and B2 fractures. Did the patients perform preoperative CT in all the patients? We cannot rely only on radiographs; intraoperative evaluation is more valuable.

  1. Did the groups receive a comparable number of MIPO or ORIF surgery? Please explain.

This paper forms part of the growing knowledge of the differences in outcomes of different patterns of periprosthetic fractures. A major revision is needed.

Author Response

Dear. Reviewer 2

Thank you for your sincere response.

According to your comments, we did our best to revise our paper. We believe that this process makes our study more valuable. Our answers to your comments are below.

<Response to Reviewer 2 Comments>

How was the sample size of patients calculated? Is this sample size enough to compare the groups and generalize the results? It is not clear. If the sample is not the appropriate one, the results are invalid. Did the number of patients comparable between groups? (6, 7,36 and 12) à Thanks for your comment. I agree with your point. However, it is extremely rare for periprosthetic fractures to have a transverse pattern. Our study has 7 cases, which alone may be difficult to apply to all common cases. However, compared to other fracture patterns, we experienced treatment failures in an absolute greater number, so we started this study, and it can be said that meaningful results were obtained. And this study is meaningful because we have succeeded in reoperation using an additional dual plate or using strut allograft for this fracture type failure.

Did all the patients undergo a primary uneventful total hip arthroplasty before periprosthetic fracture? Did all the patients receive the same or at least primary stems? It seems to me that the primary THA in the patient of group B in Figure 1 was not uneventful (intraoperative fracture? / this is not a primary stem/ proximal osteolysis)

à Thanks for your comment. We added that contents by table 4.

Table 3: In my opinion, the paper's main disadvantage is the differential diagnosis between B1 and B2 fractures. Did the patients perform preoperative CT in all the patients? We cannot rely only on radiographs; intraoperative evaluation is more valuable.

à Thanks for your comment. We generally do not judge B1 and B2 during surgery. Because the direction of treatment is completely different, it is a principle to judge the two types before surgery. As you mentioned, I think there are some cases where there is no clear distinction between B1 and B2. In this study, stem subsidence of 2 patients occurred in group C, and the results were not different even if these cases were excluded. And Cases where the distinction between B1 and B2 fractures was not clear were excluded from this study. I will add more information on this part.

Did the groups receive a comparable number of MIPO or ORIF surgery? Please explain.

à Thanks for your comment. We added that contents by table 5

Round 2

Reviewer 2 Report

I want to thank the authors for the time they spent answering the questions.

I believe that all these remarks 1. sample size 2. different size between groups  3. inability to differentiate between B1 and B2 fractures preoperatively must be included as limitations of the study. 

Author Response

Dear. Reviewer 2

Thank you for your sincere response.

According to your comments, we did our best to revise our paper. We believe that this process makes our study more valuable. Our answers to your comments are below.

<Response to Reviewer 2 Comments>

I believe that all these remarks 1. sample size 2. different size between groups  3. inability to differentiate between B1 and B2 fractures preoperatively must be included as limitations of the study.

-> Thanks for your comment. I added your advice at limitation part in Discussion portion.
